# microRNAs Control Antiviral Immune Response, Cell Death and Chemotaxis Pathways in Human Neuronal Precursor Cells (NPCs) during Zika Virus Infection

**DOI:** 10.3390/ijms231810282

**Published:** 2022-09-07

**Authors:** Carolina M. Polonio, Patrick da Silva, Fabiele B. Russo, Brendo R. N. Hyppolito, Nagela G. Zanluqui, Cecília Benazzato, Patrícia C. B. Beltrão-Braga, Sandra M. Muxel, Jean Pierre S. Peron

**Affiliations:** 1Neuroimmune Interactions Laboratory, Department of Immunology, University of São Paulo, São Paulo 05508-000, Brazil; 2Scientific Platform Pasteur-USP (SPPU), University of São Paulo, São Paulo 05508-000, Brazil; 3Disease Modeling Laboratory at Department of Microbiology, Institute of Biomedical Sciences, São Paulo 05508-000, Brazil; 4Immunopathology and Allergy Post Graduate Program, School of Medicine, University of São Paulo, São Paulo 05508-000, Brazil

**Keywords:** microRNA, ZIKV, inflammation, cell death, chemotaxis

## Abstract

Viral infections have always been a serious burden to public health, increasing morbidity and mortality rates worldwide. Zika virus (ZIKV) is a flavivirus transmitted by the *Aedes aegypti* vector and the causative agent of severe fetal neuropathogenesis and microcephaly. The virus crosses the placenta and reaches the fetal brain, mainly causing the death of neuronal precursor cells (NPCs), glial inflammation, and subsequent tissue damage. Genetic differences, mainly related to the antiviral immune response and cell death pathways greatly influence the susceptibility to infection. These components are modulated by many factors, including microRNAs (miRNAs). MiRNAs are small noncoding RNAs that regulate post-transcriptionally the overall gene expression, including genes for the neurodevelopment and the formation of neural circuits. In this context, we investigated the pathways and target genes of miRNAs modulated in NPCs infected with ZIKV. We observed downregulation of miR-302b, miR-302c and miR-194, whereas miR-30c was upregulated in ZIKV infected human NPCs in vitro. The analysis of a public dataset of ZIKV-infected human NPCs evidenced 262 upregulated and 3 downregulated genes, of which 142 were the target of the aforementioned miRNAs. Further, we confirmed a correlation between miRNA and target genes affecting pathways related to antiviral immune response, cell death and immune cells chemotaxis, all of which could contribute to the establishment of microcephaly and brain lesions. Here, we suggest that miRNAs target gene expression in infected NPCs, directly contributing to the pathogenesis of fetal microcephaly.

## 1. Introduction

The Zika virus (ZIKV) is an arbovirus from the *Flaviviridae* family with a symmetrical structure, icosahedral nucleocapsid, enveloped and a single-stranded positively oriented RNA genome. It was first isolated in 1947 from the blood of *Rhesus* sp. sentinel monkeys in the Ziika forest in Uganda, Africa. Since then, the clinical relevance was negligible, as it rarely caused human infections in Africa and Asia [1].

However, in the first half of 2015, a generalized epidemic of ZIKV erupted in South and Central America, being the northeast of Brazil the most affected region. Clinicians have noticed a significant increase in the number of serious neurological complications cases, mostly Guillain-Barré Syndrome and babies born with microcephaly. Along with microcephaly, newborns may also display other severe features such as brain calcifications, retinal damage, growth restriction, and, sometimes, arthrogryposis, characterizing the Congenital ZIKV Syndrome (CZS) [2,3,4,5].

One of the first pieces of evidence of a causal relationship between ZIKV and microcephaly was demonstrated by a case report showing intrauterine growth restriction, cortical calcifications, a brain circumference of 26 cm and the presence of ZIKV in the fetal brain [6]. Subsequently, other studies confirmed the causal relationship between ZIKV and microcephaly using experimental models in vitro and in vivo [7,8,9].

The results clearly demonstrate that the virus has a tropism for the central nervous system of fetuses, resulting in the destruction of neuronal precursor cells (NPCs). Conversely, through intracerebroventricular viral injection at E13.5, it was demonstrated that the ventricular and subventricular areas were infected and damaged. A reduction in TBR1^+^, SOX2^+^ and FOXP2^+^ cells was observed, associated with a decreased cortical plate, ventricular zone and subventricular zone brain thickness, resulting in reduced brain size [8]. In addition, we have previously demonstrated that infected SJL pregnant mice had intrauterine growth restriction and decreased cranial measurements with smaller cerebral cortex [9]. In fact, ZIKV infects NPCs and neurons, significantly upregulating autophagy and apoptosis, evidenced by TUNEL and active caspase-3 staining. Furthermore, a reduction in the cortical layer was observed due to the death of TUJ^+^, SOX-2^+^ and TBR-1^+^ NPCs, evidenced by in vitro studies using mini-brains, also corroborated by other groups [9,10,11]. It was further corroborated using monkeys that the decreased cortical layer is caused by apoptosis of NPCs in the fetuses [12,13].

Although the neurological changes caused by ZIKV lead to irreversible damage to infected fetuses and neonates, it is known that only 6 to 12% of infected pregnant women generate children who develop CZS [14,15]. Post-transcriptional mechanisms that regulate the antiviral immune response, chemotaxis of immune cells and cell death seem to be important during ZIKV infection. In this context, miRNAs may play a fundamental role, but very little is known about their participation in the mechanisms underlying ZIKV neuropathogenesis.

MiRNAs are small non-coding single-stranded RNAs that perform post-transcriptional regulation of gene expression by binding to the 3′ untranslated region and thus destabilizing messenger RNAs (mRNAs) of several genes and then abrogating protein translation [16]. The regulatory capacity of miRNAs and its study has gained notoriety over the years, since they participate in different diseases, such as cancer [17], diabetes [18], multiple sclerosis [19] and viral infections, including neurotrophic flavivirus [20,21]. MiRNA expression is well controlled in physiological conditions and their deregulation is associated with pathologies, including those associated with neurological development [22,23]. Interestingly, Dicer-deficient mice, an important molecule for miRNAs biogenesis, suffer spontaneous abortion [24]. The miR-9 is specifically expressed by NPCs during neuronal differentiation, and miR-9-deficient mice showed ventriculomegaly and decreased cortex thickness when compared to WT mice, showing the miRNAs important during embryonic development [25]. In addition, some miRNAs can participate in the regulation of cellular processes, such as apoptosis, antiviral immune response, and chemotaxis of immune cells. Other flaviviruses, such as the Japanese Encephalitis Virus and West Nile Virus, produce or can induce the production of miRNAs that play a fundamental role in their replicative cycle or inhibit the immune response, and favor infection [26,27]. Japanese Encephalitis Virus increases miR-146 expression to negatively regulate TRAF6, IRAK1 and STAT1 molecules, suppressing NF-κB activation and regulating Interferon Stimulated Genes (ISG) to facilitate viral replication in human microglial cells [26]. Moreover, in Neuro-2A, a mouse neuroblast cell line, Japanese Encephalitis Virus or West Nile Virus infection upregulates miR-451a to decrease 14-3-3ζ, increasing the phosphorylation of JNK protein and leading to apoptosis [28].

In this context, miRNAs are important under many different scenarios affecting several important pathways, such as for the neurodevelopment or local immune response. Here, we evaluated the miRNA profile of human NPCs infected with ZIKV for further comparison with the transcriptional profile of NPCs from a public dataset. This associated analysis demonstrated that ZIKV regulates the expression of miRNAs targeting genes of the antiviral immune response, immune cell chemotaxis and cell death in NPCs, many of which were confirmed to be modulated in the public dataset evaluated. Thus, we propose here that a dysregulated miRNA profile may correlate with the establishment of microcephaly and brain lesions during ZIKV infection. This may improve the understanding of the mechanism of neuropathogenesis and may favor the identification of promising target genes for possible therapeutic interventions to reduce the overall impact on brain pathology.

## 2. Results

### 2.1. ZIKV Infects Human NPCs

Firstly, we showed that human induced pluripotent stem cells (iPSC)-derived NPCs from the dental pulp of a healthy single individual were infected by ZIKV. After 48 h of infection, we observed an increase in cell death (Figure 1A) characterized by cellular morphology. It is worth mentioning that we decided to use 48 h timepoint, as in our previous work we have performed a detailed kinetic [9]. We observed that ZIKV replicates in NPCs reaching the peak of viral replication and cell death at 96 h. Because of the high cell death rate, we decided to use in the present study the 48 h timepoint. Next, we confirmed the ability of ZIKV to infect NPCs by detecting viral RNA by qPCR (Figure 1B), infective particles by plaque-forming unit (PFU) (Figure 1C) and the presence of viral envelope by immunofluorescence (Figure 1D). Then, we verified the expression of characteristic markers of NPCs, such as Musashi-1, Nestin, SOX2 and the absence of MAP2 (Appendix A).

### 2.2. ZIKV Modulates miRNAs Profile in Human NPCs

By comparing the miRNA profile of infected (ZIKV^BR^) and uninfected (CTRL) human NPCs (Figure 2A), and using a cut-off of 2 folds, we observed that three miRNAs were downregulated (Figure 2A,B) by ZIKV infection, being miR-302b, miR-302c, both from the same family, and miR-194, whereas miR-30c was upregulated (Figure 2A,C).

In order to deepen our analyses, we analyzed a public RNA-seq dataset from human induced-NPCs (hiNPCs) after 48 h of ZIKV Brazilian strain infection (NCBI BioProject PRJNA551246) [29], a very similar model to ours. Thus, we searched for genes upregulated by ZIKV in RNA-seq that were putative targets of the downregulated miRNAs miR-302b, miR-302c and miR-194, and for downregulated genes as putative targets of the miR-30c upregulated by infection. We performed the prediction of mRNA targets using the miRWalk 2.0 platform comparing three databases: TargetScan, Probability of Interaction by Target Accessibility (PITA), and miRanda, using the Bonferroni statistical test and considering the genes statistically modulated when the *p*-value is <0.05. Then, we designed the interactome of modulated miRNAs and their predicted targets (Figure 2D). As expected, several predicted target genes were upregulated when miRNAs were downregulated in ZIKV-infected hiNPCs. These genes are involved in pathways related to cell death, chemotaxis, antiviral immune response, type I IFN regulation, primary autosomal recessive microcephaly, neurodevelopment, antigen processing and presentation, and others (Figure 2D). Thus, we can speculate that ZIKV dysregulates the miRNA network, and consequently, impact the putative mRNA target genes of human NPCs.

### 2.3. ZIKV Downregulates miRNAs Increasing Cell Death of Human NPCs

Next, we investigated genes involved in cell death that could be putative targets of miR-302b, miR-302c and miR-194 in ZIKV-infected hiNPCs (NCBI BioProject PRJNA551246). We found increased levels of *NLRC5* (NLR Family CARD Domain Containing 5), a molecule that acts both as a transcriptional activator of MHCI, and also a Pattern Recognition Receptor (PRRs) already shown to be important for ant-viral responses [30], and *CASP1* (Caspase-1), which together can induce cell death by pyroptosis [31]. Other genes involved in cell death pathways were also upregulated by ZIKV, such as: *TLR3*, *CD274*, *TNFSF10*, *GBP1*, *PMAIP1*, *PDC1LG2*, and *MLKL* (Figure 3A,B). Moreover, gene ontology (GO) analysis showed an increase in TLR3 signaling, cell death, and IL-1β signaling and production pathways (Figure 3C), which was further corroborated by increased IL-1β production on the supernatants of human NPCs infected with ZIKV (Figure 3D). These data may suggest a modulation of miRNA-mRNA targeting the inflammatory cell death pathway during ZIKV infection.

### 2.4. ZIKV Downregulates miRNAs to Recruit Immune Cells and Modulate the Immune Response

Next, we searched for genes from the immune response that can also be targets of the downregulated miRNAs in ZIKV-infected hiNPCs (NCBI BioProject PRJNA551246). The RNA-seq analysis showed an upregulation of chemokines genes, such as, *CXCL3*, *CXCL9*, *CXCL10*, and *IL15*, also predicted targets of downregulated miRNAs (Figure 4A,B). GO analysis confirmed the regulation of cell migration and chemotaxis (Figure 4C). Hence, we observed increased IL-8 production on the supernatant of human NPCs infected with ZIKV (Figure 4D), which suggests an increase in neutrophil recruitment [32].

As expected, ZIKV enhanced type I interferon (IFN) signaling, evidenced by the increased expression of Interferon Stimulated Genes (ISG): *ISG20*, *IFI6*, *MX1*, *MX2*, *OAS1*, *OAS2*, *OAS3*, *OASL*, *IFIT1*, *IFIT3*, *IFIT15*, *IFI44*, *IFIH1*, *IFI44L*, *IFI27*, and *PARP12* (Figure 5A,B). Accordingly, our data demonstrated an increased *MXD1* gene expression by ZIKV infection, a transcriptional repressor of Myc transcriptional activity [33] and a target of all downregulated miRNAs. In addition, the *ISG20* gene was also upregulated in NPCs 48 h post-infection (Figure 5D). Molecules involved in type I IFN signaling are also increased by ZIKV infection, such as *STAT1*, *DDX60L*, *DDX60*, *DDX58*, *STAT2*, and *IRF7* (Figure 5A,B). Conversely, *RTP4* gene expression was upregulated, an inhibitor of type I IFN responses (Figure 5A,B). Curiously, GO analysis showed type I IFN signaling enriched, along with increased negative regulation of this pathway (Figure 5C). Moreover, the upregulated miR-30c can target the cytokines TNF-α and IL-6. Interestingly, there is no difference in TNF-α and IL-6 production, which may suggest that miR-30c was not allowing the increase in these cytokines during ZIKV infection (Figure 5E). The data suggest that the immune system is trying to eliminate the virus, at the same time the ZIKV is using miRNAs to escape from the immune system.

### 2.5. ZIKV Downregulates miRNAs to Increase Ubiquitination, and Antigen Presentation

Antigen presentation through MHCI is classically important during the antiviral immune response. This pathway is dependent on protein degradation by the ubiquitin-proteosome machinery. Here, we also evaluated the public data sets whether genes related to this phenomenon are also modulated. Interestingly, ubiquitination-related genes were upregulated in the RNA-seq of ZIKV-infected hiNPCs, including *TRIM38*, *HERC5*, *HERC6*, *UBA7*, *PSMB8-1*, *PSMB9-5*, *UBE2L6*, and *NEURL3*. In addition, genes involved in antigen processing and presentation, namely *TAP-2*, *HLA-E*, and *CTSS*, were up-regulated, whereas putative miRNAs were downregulated (Figure 6A,B). Hence, pathways associated with antigen processing and presentation via MHC I, and ubiquitination were enhanced by ZIKV infection (Figure 6C).

## 3. Discussion

The importance of miRNAs during flavivirus infection, specifically ZIKV, has gained notoriety in recent years, especially because miRNAs from viruses or hosts can regulate resistance and susceptibility to infection. Interestingly, there are 47 miRNAs encoded in the ZIKV genome that target pathways involved in cell signaling regulation, neurological functions, and fetal development [34]. Furthermore, the E protein from ZIKV modulates miRNA expression to regulate cell cycle and developmental processes [35]. Moreover, ZIKV capsid directly suppresses Dicer activity in neural stem cells, and, consequently, decreasing miRNA biogenesis, impacting corticogenesis [36]. In association, neural stem cells produce extracellular vesicles containing miRNAs, leading to cell-cell communication and impacting on neurodevelopment and oxidative stress [37]. Therefore, understanding the role of miRNAs in highly susceptible CNS cell populations to ZIKV infection is extremely relevant. In this sense, here we performed a hybrid study, in which we evaluated the miRNA expression profile of human ZIKV-infected NPCs and compared it with the transcriptomics of a public data set, especially focusing on the miRNA target genes and GOs pathways associated.

It is worth mentioning that the microRNAs not only destabilize and degrade mRNAs, but can also inhibit protein translation and perform transcriptional silencing by competing for the 5′CAP, preventing the association of mRNAs with ribosomes [38], or by blocking the initiation of translation, preventing the interaction between poly-A tail and poly-A binding protein C1 [39]. In addition, miRNAs can perform transcriptional silencing that involves RNA processing bodies (p-bodies), which are cytoplasmic ribonucleoprotein aggregates. MiRNAs direct target mRNAs to p-bodies, where they are temporarily and reversibly repressed or destabilized [40]. Although there are others regulation mechanisms by miRNAs, the most common is still transcriptional repression, and it is this function that we evaluated in this work.

Here we observed the downregulation of miR-302b, miR-302c, and miR-194, and the upregulation of miR-30c in ZIKV-infected hNPCs. The miRNAs dysregulation correlated with GOs pathways that may promote cell death and immune cells chemotaxis, which may be harmful to the developing brain. The miR-302b and miR-302c encompass the miR-302–367 cluster, highly conserved in vertebrates, sharing the seed sequence and mRNA targets [41]. They have demonstrated function in cell proliferation and differentiation of human embryogenic stem cells. Interestingly, factors implicated in human embryogenic stem cell differentiation such as Nanog, Oct3/4, Rex1 and Sox2, upregulate this cluster expression [42]. The miR-302 suppresses AKT1 translation and increases the pluripotent factor OCT4 in hiPSCs [43]. The overexpression of miR-302–367 cluster reprograms astrocytes to neuroblasts and then in neurons in vitro and *in vivo*, and induces behavioral improvement and neural repair in the murine model of Alzheimer’s disease [44]. Overexpression of the miR-302 family reprograms human hair follicle cells increasing Nanog, Oct3/4, and Sox2 expression [45]. The miR-194 is located in the miR194–192 cluster within the human genome and can be induced by hepatocyte nuclear factor-1a [46]. Primary neurons express miR-194 [47] which regulates the proliferation and apoptosis of hippocampal neurons [48] and neuronal morphogenesis [49].

On the other hand, miR-30c expression is associated with both pathological and neuroprotective functions [50]. miR-30c was first described in brain tissues [51] and is encoded with the miR-30 family (miR-30a-d), playing a regulatory function in tissue development [52]. This miRNA family was already demonstrated to be involved in the proliferation and differentiation of murine NPCs impacting neurogenesis [53].

It is known that NPCs are the most affected cells by ZIKV in the CNS. Our data confirmed the enrichment of cell death pathways during ZIKV infection, as well as of IL-1β cytokine signaling pathways, also evidencing death by pyroptosis. It has already been demonstrated that ZIKV induces pyroptosis either dependent [54] or independent [55] of caspase-1 in NPCs. Furthermore, our data show a significant increase in the NLRC5 expression, a subgroup of NOD-like receptors, which facilitates caspase-1 production leading to IL-1β secretion [56]. The miR-302b and miR-194 can regulate IL-1β [57]. Other cell death pathways caused by ZIKV, and previously described in NPCs, are apoptosis and autophagy, demonstrated by the upregulation of *Bmf*, *Irfm1*, *Bcl2*, *Htt*, *Casp6* and *Abl19* in the brain of SJL susceptible mice. We also observed an upregulation of Toll-like receptor 3 (TLR3) signaling, which can lead to apoptosis of NPCs-derived brain organoids, and consequently attenuation of neurogenesis [58].

Adult patients in the acute phase of ZIKV infection produce high levels of chemokines such as *CXCL3*, *CXCL9*, *CXCL10* and *IL15* in the serum [59], as well as microcephaly babies in the cerebrospinal fluid [29], especially when developing neurological symptoms. This is also in agreement with our analysis of the public data set of NPCs transcriptomics, where GOs pathways related to cell migration and chemotaxis were enriched. Conversely, mir-302c was downregulated in our samples, which was also associated with increased secretion of IL-8 in the NPCs supernatant, a known neutrophil chemokine. In fact, we have recently demonstrated that ZIKV infects neutrophils without interfering with cell viability, reactive oxidative species production, and neutrophil extracellular traps release, suggesting a role in disseminating infection throughout the body, especially to the placenta [60]. In addition, patients infected by ZIKV have higher production of IL-8 cytokine than Chikungunya-infected patients [60]. These chemokines are also known to attract CD8^+^ T cells [61], which migrate to the CNS and exert their cytotoxicity functions [62]. Although that is good for viral replication control, also can lead to the death of infected cells.

Curiously, besides an important innate immunity-related molecule, NLRC5 is also an MHC class I transactivator, as it associates with and activates the promoters of MHC I, beta2-microglobulin, and transporters and proteases associated with antigen processing and presentation [63,64,65]. This leads us to believe that NLRC5, besides inducing pyroptotic cell death of NPCs, can also assist in the activation of CD8^+^ T cells via antigen presentation by MHC class I, leading to the killing of infected cells.

The role of miRNAs during flavivirus infections has been ambiguously described. Some flaviviruses modulate miRNAs expression to assist in viral replication and cell death. For example, miR-29c favors Porcine Reproductive and Respiratory Syndrome Virus replication in alveolar macrophages [66], whereas miR-21 promotes Dengue virus replication in human liver carcinoma cells [67]. On the Other hand, other miRNAs favor the antiviral immune response preventing virus spread. For example, the Japanese Encephalitis Virus increases miR-15b expression, triggering Ring Finger Protein 125 inhibition, a negative regulator of RIG-I signaling, and promoting inflammatory response, including type I IFNs production, and a decrease in glial activation and neuronal damage [68]. Corroborating this, we showed this ambiguous role of miRNAs by demonstrating an increase in type I IFNs signaling characterized by an upregulation of several ISGs and signaling molecules, at the same time of increased antiviral immune response inhibitor, such as Receptor Transporter Protein 4. Moreover, the fact that the pro-inflammatory cytokines, such as TNF-α and IL-6 were not highly increased in the supernatant of the infected NPCs may also suggest a regulation through miRNA. For example, the functional inhibition of miR-302 increases *Tnf* mRNA levels in macrophages [69]. Inhibition of miR-30e leads to NF-κB activation by targeting IkBα 3′ untranslated region, inducing IFN-β and suppressing dengue virus replication [70].

Another mechanism that assists in viral replication is polyubiquitination of E protein from ZIKV. It has been shown that the ZIKV E protein is polyubiquitinated by E3 ubiquitin ligase TRIM7. Curiously, ZIKV replicates less in TRIM7^−/−^ mice [71]. Confirming this, our data set analysis showed an upregulation of different genes associated with ubiquitination, many of which correlated with the miRNAs.

In summary, here we have identified the ZIKV’s ability to modulate miRNAs expression in human NPCs, either directly or indirectly. We observed that many of these miRNAs target important pathways, including cell death, immune cell recruitment, regulation of immune response, ubiquitination and antigen processing and presentation, probably to favor viral replication and success of infection. However, this seems to be deleterious to the brain tissue. Especially for the fetal developing brain, as NPCs generate many glial and neuronal cells. Corroborating our findings, the transcriptional profile of a public dataset of infected human NPCs had many of the miRNA target genes modulated accordingly. Our goal here was to call attention to the importance of the miRNA network for the neuropathogenesis of ZIKV congenital syndrome. Our findings concerning miRNAs expression changes in association with the bioinformatics analysis of a public data set corroborates this idea. We believe our work contributes to a better understanding of ZIKV infection pathology and neuropathology and may point to miRNA as targets for possible therapeutic intervention.

## 4. Material and Methods

### 4.1. Human NPCs Culture

We used a human iPSC clone derived from the human dental pulp of a single control individual (CAAE 58219416.0.0000.5467) that was previously characterized in the laboratory of Beltrão-Braga [72]. The cell line tested negative for mycoplasma contamination. Briefly, high passage iPSC colonies were plated for 5 days with mTSeR media (Stem cell Technologies, Vancouver, BC, Canada, 85850). On the fifth day, media was changed to N2 media (DMEM/F12 media (Gibco, Itapevi, SP, Brazil, 10565018), supplemented with 1× N2 supplement (Gibco, Itapevi, SP, Brazil, 17502048), 1 μM dorsomorphin (Stemgent, San Diego, CA, USA, 04-0024), and 1 μM SB431542 (Stemgent, San Diego, CA, USA, 04-0010)) for 48 h. Colonies were separated from the plate and cultured in suspension as embryonic bodies (eBs) for 5 days at 90 RPM in N2 media. eBs were plated on matrigell coated plates with NBF media (DMEM/F12 media supplemented with 0.5× N2, 0.5× B27 (Gibco, Itapevi, SP, Brazil, 17504044), 20 ng/mL FGF2 (Gibco, Itapevi, SP, Brazil, 13256-029) and 1% penicillin/streptomycin (Gibco, Itapevi, SP, Brazil, 15140-122)). The emerged rosettes containing the NPCs were manually harvested, dissociated and plated in a double-coat with a 10 μg/mL polyornithine solution (Sigma-Aldrich, San Louis, MO, USA, 27378-49-0), and 2.5 μg/mL laminin (Gibco, Itapevi, SP, Brazil, A29248). The NPC population was expanded using NBF media. All experiments were performed with the Ethics Committee approval of the Institute of Biomedical Sciences, protocol number 1001.

### 4.2. Viral Culture and Amplification

A lyophilized isolate of ZIKV from a clinical case in Brazil (ZIKV^BR^) (BeH823339), provided by Instituto Evandro Chagas in Belém—Pará, was reconstituted in 0.5 mL of sterile DEPC water (Invitrogen, Itapevi, SP, Brazil, AM9915G). *Aedes albopictus* mosquito cells (C6/36) were previously prepared for virus culture and maintained in L-15 media (Sigma-Aldrich, San Louis, MO, USA, L1518) supplemented with 10% fetal bovine serum (FBS—Gibco, Itapevi, SP, Brazil, 12657–029), 1% non-essential amino acids (LGC, Sao Paulo, SP, Brazil, BR30238–01), 1% Sodium Pyruvate (LGC, Sao Paulo, SP, Brazil, BR30239–01), 1% Penicillin/Streptomycin, 0.05% Amphotericin B (Gibco, Itapevi, SP, Brazil, 15290026) at 27 °C in the absence of CO_2_. Further, it was produced the first subculture (T1) by inoculating 50 μL of the viral sample plus 900 μL of incomplete L-15 media into C6/36 cells seeded in 25 cm^2^ flasks with approximately 70% confluent monolayer for one hour, with gentle agitation every 10 min to allow homogeneous adsorption of the viruses. At the end of the adsorption period, we added 5 mL of the L-15 culture media, with 2% FBS, 1% non-essential amino acids and 1% sodium pyruvate. After four days, the supernatant was collected. The second subculture (T2) was made blindly by transferring 500 μL of the T1 supernatant plus 4,5 mL of incomplete L-15 media into C6/36 cells in 182 cm^2^ flasks for more than four days, and the supernatant was collected. The third subculture (T3) was made blindly by transferring 500 μL of the T2 supernatant plus 4,5 mL of incomplete L-15 media into C6/36 cells in 182 cm^2^ flasks for eight days when cells presented morphological changes. The T1, T2 and T3 viral stock were produced by Prof. Edson Durigon (Biomedical Sciences Institute, University of São Paulo). From T3, we performed virus stocks by inoculating C6/36 cells seeded in 182 cm^2^ flasks with 500 μL of T3 stock virus plus 4,5 mL of incomplete L-15 media for 1 h at 27 °C. Further, we added 20 mL of supplemented L-15 media with 2% FBS and 1% Penicillin/Streptomycin. The cells were incubated for 4 days until 70% of cytopathic effect. Supernatants were harvested, centrifuged for 5 min at 10.000× *g*, 4 °C, concentrated using 5× Polyethylene Glycol (PEG, Sigma-Aldrich, San Louis, MO, USA, MAK343), titrated, and snap-frozen at −80 °C until use.

### 4.3. Viral Titration

The titer (in PFU/mL) of each subculture was obtained by plaque forming unit (PFU) assay to determine the amount of released infectious viral particles. Briefly, 1 × 10^5^ Vero CCL81 cells were seeded in a 24-wells plate, then a serial dilution of the ZIKV^BR^ viral stock was performed from 10^−1^ to 10^−11^ using only 200 μL of each dilution in each well during 1 h at 37 °C, 5% CO_2_ to allow virus adsorption. Subsequently, each well was covered with complete 3% carboxymethylcellulose (CMC) media (Sigma-Aldrich, San Louis, MO, USA, C4888) diluted in L-15 media supplemented with 2% FBS in 1:1 proportion. After five days of incubation at 37 °C and 5% CO_2_, visualization of plates was performed using a crystal violet solution. The viral titer was determined by the division of the average number of plaques for dilution and the total dilution factor. $PFU=\frac{\mathrm{average}\;\mathrm{of}\;\mathrm{plaques}}{\mathrm{dilution}\;\times \;\mathrm{volume}}$.

Viral copies were determined by real-time PCR. Total RNA was extracted with Trizol^®^ (Invitrogen, Itapevi, SP, Brazil, 15596026). The concentration of mRNA was measured by spectrophotometer according to 260/230 and 260/280 purity ratios.

### 4.4. In Vitro Infection

The human NPCs were seeded in a 24-wells plate and infected or not with ZIKV^BR^ (BeH823339). Viral samples were diluted to MOI = 1 and added to the cells. For viral adsorption, monolayer cells were incubated for 1 h at 37 °C and 5% CO_2_. Then, the inoculum was removed and cells were washed once with 1× PBS. We added specific culture media, which were subsequently incubated at 37 °C and 5% CO_2_ for 48 h after infection.

### 4.5. RNA Extraction and cDNA Synthesis

Human NPCs were lysed with 1 mL of Trizol reagent for 5 min at room temperature. Next, 200 μL of chloroform (Synth, Diadema, SP, Brazil, 00C1062.06 BJ) was added to separate RNA, DNA and proteins, after centrifugation at 12,000× *g* for 15 min. The transparent upper portion containing RNA was carefully removed and transferred to another tube to which 500 μL of isopropanol (Synth, Diadema, SP, Brazil, A1078.01.BJ) was added. The mixture was kept at room temperature for 10 min and centrifuged at 12,000× *g* for 10 min and the supernatant was carefully removed. The pellet was washed with 1 mL 75% ethanol and centrifuged at 7600× *g* for 5 min at 4 °C 3 times. Finally, RNA was reconstituted in DEPC water. The concentration of purified total RNA was determined in a spectrophotometer at 260/280 nm (NanoDrop 2000, ThermoFisher, Waltham, MA, USA, ND-2000). For cDNA synthesis, a reverse transcription reaction was performed from 2 μg purified total RNA using High-Capacity cDNA Reverse Transcription Kit (Applied Biosystems, Waltham, MA, USA, 4368814), according to manufacturer’s instructions. The mixture was taken to the thermocycler (QuantStudio3, Applied Biosystems, Waltham, MA, USA) and subjected to the following cycles: 25 °C for 10′; 37 °C for 120′; and 85 °C for 5′. Further, the samples were diluted for 10 ng/μL concentration.

### 4.6. Real-Time PCR Quantification

From the cDNA, the mRNA expression was evaluated by real-time PCR (qPCR). For each PCR reaction, 200 μM forward primer, and 200 μM reverse primer was mixed with 5 μL of SYBR Green Master Mix (Applied Biosystems, Waltham, MA, USA, 4367659). A total of 46 ng of cDNA sample was added to this solution. The solutions were taken to the QuantStudio3 device (Applied Biosystems, Waltham, MA, USA) and subjected to different stages (1—50 °C for 2 min; 2—95 °C for 10 min; 3—95 °C for 15 s and 4—60 °C for 1 min. Steps 3 and 4 were repeated 4× for genes and 45× for ZIKV). The median cycle threshold (C_t_) value from experimental replicates and the 2^−ΔΔCt^ method was used for relative quantification analysis and all C_t_ values were normalized by constitutive gene HPRT. Below is the list of primers (Table 1), which were used in real-time PCR experiments:

### 4.7. miRNAs Expression

From total RNA, 2 μg was subjected to reverse transcription reaction only for mature miRNAs using miScript II RT kit (Qiagen, Hilden, Germany, 218161). For that, 12 μL of RNA were added to 8 μL of mix (4 μL of 5× miScript HiSpecBuffer, 2 μL of 10× miScript Nucleics Mix and 2 μL of miScrpt Reverse Transcritase mix). The mixture was taken to a thermocycler (QuantStudio 3, Applied Biosystems, Waltham, MA, USA), subjected to the following cycles: 37 °C for 60′; 95 °C for 5′, and immediately placed on ice. A total of 90 μL of water was added to this mixture and further, 100 μL of the synthesized cDNA was mixed with 1375 μL of 2× QuantiTect SYBR Green PCR, 275 μL 10× miScript Universal Primer (Qiagen, Hilden, Germany, 218075). Then, 25 μL of this solution was added to each well of a 96-well miScript miRNA PCR Array miFinder plate (Qiagen, Hilden, Germany —Human MIHS-001Z), containing a primer for each miRNA in each well. In sequence, this plate was taken to the thermocycler (QuantStudio3, Applied Biosystems, Waltham, MA, USA) and subjected to the following cycles: 95 °C for 15″; 94 °C for 15″; 55 °C for 30″ and 70 °C for 30″ × 40 repetitions. The curves were normalized by the expression of 8 endogenous genes. Analyses were conducted through Qiagen’s analytics platform: https://www.qiagen.com/br/shop/genes-and-pathways/technology-portals/browse-qpcr/qrt-pcr-for-mrna-expression/data-analysis/~/link.aspx?_id=93C80B99536C4FCDBD1E5FC0759F1324&_z=z (accessed on 15 September 2019).

### 4.8. miRNAs Computational Analysis

To predict the possible mRNA targets modulated by the miRNAs regulated during ZIKV infection, we used the mirWalk 2.0, a free platform that compares analysis from other sites (PITA—Probability of Interaction by Target Accessibility, TargetScan and miRanda).

### 4.9. Mapping of RNA-Seq Libraries and Differential Gene Expression Analysis

RNA-seq raw data were obtained from human induced-neuroprogenitor stem cells (hiNPCs) infected with ZIKV, available at NCBI BioProject under accession number PRJNA551246. For the purposes of this study, we selected and compared hiNPCs infected with PE243 ZIKV strain and uninfected control. Initially, raw reads served as input for Trimmomatic 0.40 [73], which performed quality filtering removing Illumina adaptor sequences, low-quality bases (phred score quality > 20), and short reads (PE-phred33 ILLUMINACLIP:truseq.fa:2:30:10 LEADING:3 TRAILING:3 SLIDINGWINDOW:30:20 MINLEN:36). Trimming was followed by read error correction by SGA *k*-mer-based algorithm [74] version 0.9.9 with standard settings. Next the reads were mapped against the Genome Reference Consortium Human Build 38 (GRCh38) using HISAT2 software [75] following previously described optimization [76] (--rna-strandness RF --dta --threads 10 N 1 -L 20 -i S,1,0.5 -D 25 -R 5 --pen-noncansplice 12 --mp 1,0 --sp 3,0). The differential gene expression calling was achieved initially by counting the number of reads in each transcript through HTSeq [77] 1.99.2 with the following settings: -f bam -r pos -s no -a 10 -t exon -i gene_id -m intersection-nonempty. Finally, the count data were direct to differential analysis with DESeq2 (version 1.34.0) R package [78] with standard settings. Biological process Gene Ontology terms search was performed through the online database PANTHER with standard parameters [79].

### 4.10. Network Reconstruction

The network graph was made using the function graph_from_incidence_matrix from igraph (version 1.3.2) R 4.1.2 package. The incidence matrix was built using the information retrieved from miRNAs computational analysis performed with mirWalk 2.0. Then, it was combined with the data obtained from differential gene expression analysis and Gene Ontology, previously described, to generate the network reconstruction

### 4.11. Immunofluorescence

The human NPCs were fixed with 4% paraformaldehyde (PFA, Synth, Diadema, SP, Brazil, 01P1005.01) for 1 h at 4 °C, permeabilized and blocked with 5% BSA solution (LGC, Sao Paulo, SP, Brazil, 9048-46-8) and 0.5% TritonX-100 (LGC, Sao Paulo, SP, Brazil, 13-1315-05) for 30′ at room temperature. Primary antibodies were diluted in blocking solution and labeled overnight at 4 °C. The following day, secondary antibodies diluted in blocking solution were used for 2 h in the dark at room temperature. Finally, DAPI (Invitrogen, Itapevi, SP, Brazil, D1306) was used to label the cell nucleus at a 1:5000 dilution in 1× PBS for 10 min in the dark at room temperature. Visualization was performed using a ZOE Biorad^®^ (Hercules, CA, USA) microscope. Primary antibodies: anti-ZIKV envelope protein (rabbit, GeneTex, Irvine, CA, USA, GTX133314, 1:200); anti-MAP2 (chicken, Abcam, Cambrigde, UK, ab5392, 1:200); anti-SOX2 (goat, Abcam, Cambrigde, UK, ab97959, 1:200); anti-musashi-1 (goat, Abcam, Cambrigde, UK, ab52865, 1:200); anti-nestin (mouse, Abcam, Cambrigde, UK, ab7659, 1:200). Secondary antibodies: anti-chicken 488 (Invitrogen, Waltham, MA, USA, A-11039, 1:500); anti-rabbit 555 (Abcam, Cambrigde, UK, ab150078, 1:500); anti-goat 488 (Invitrogen, Waltham, MA, USA, A32184, 1:500); anti-goat 555 (Invitrogen, Waltham, MA, USA, A21432, 1:500).

### 4.12. Cytokine Quantification

Human NPCs supernatants were collected after 48 h post-infection and submitted to evaluation by Cytometry Beads Assay (CBA). To quantify IL-1β, Il-6, IL-8 and TNF-α cytokines, we used the CBA Human Inflammatory Cytokines Kit (BD^®^, Franklin Lakes, NJ, USA, 551811) according to the manufacturer’s instructions. Briefly, the supernatants were incubated with capture beads for 1.5 h, and washed. Next, detection beads were added and incubated for 1.5 h, and washed. The beads were acquired using the FACS Accuri C6 flow cytometry (BD Biosciences^®^, Franklin Lakes, NJ, USA) device and 2100 events were acquired.

### 4.13. Statistical Analysis

GraphPad Prism software was used to analyze all data using a t-test for simple comparations, and one-way ANOVA and a post-test of Tukey for multiple comparisons. A *p*-value < 0.05 was considered statistically significant.

## Figures and Tables

**Figure 1 ijms-23-10282-f001:**
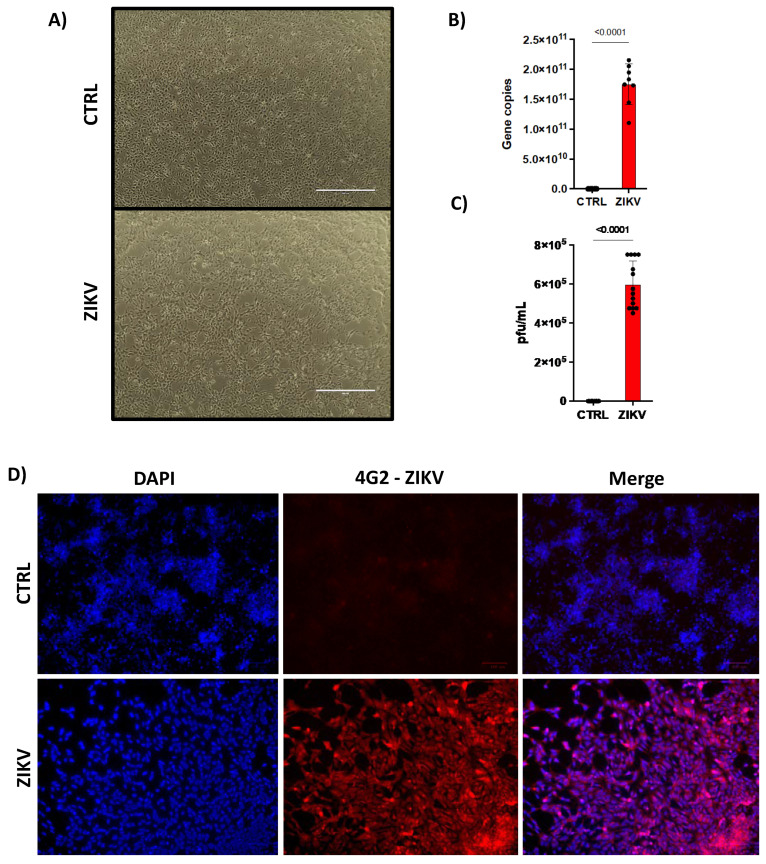
ZIKV infects human NPCs. Human NPCs were (**A**) infected or not with ZIKV (MOI = 1). 10× magnification. After 48 h post infection, they were evaluated for (**B**) viral copy numbers by qPCR (n = 14) and (**C**) PFU. (n = 8). In (**D**) ZIKV infection was confirmed by immunofluorescence with anti-envelope flavivirus antibody at 20× magnification (n = 3). Graphs represent three independent experiments. Unpaired *t*-test.

**Figure 2 ijms-23-10282-f002:**
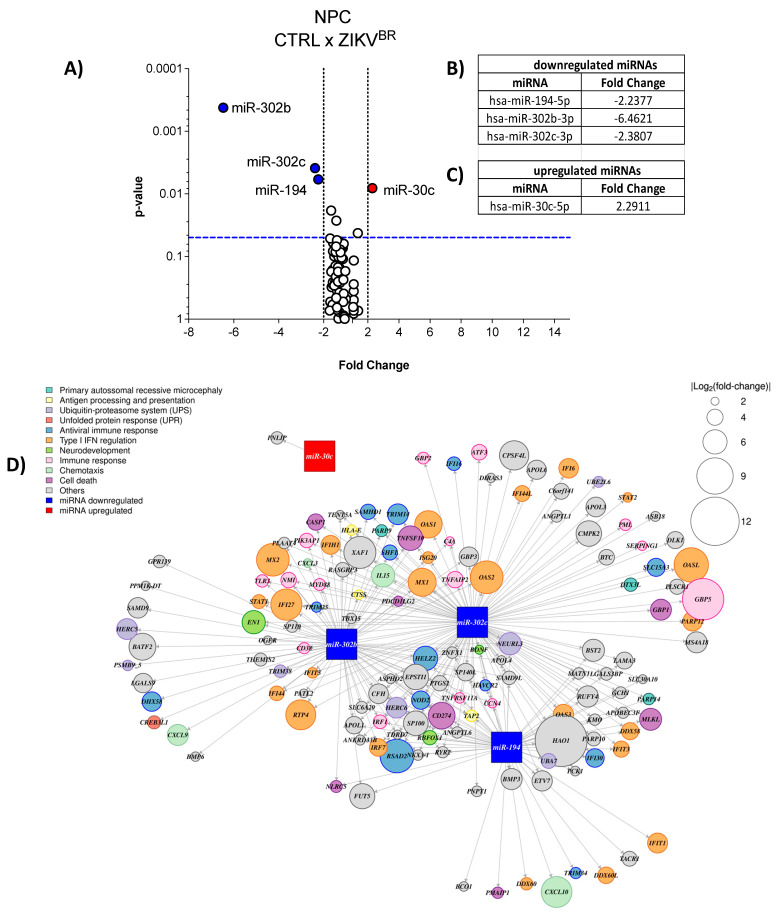
ZIKV modulates miRNA profile in human NPCs. (**A**) Human NPCs were infected or not with ZIKV (MOI = 1). After 48 h we evaluated the miRNA profile by qPCR array. Volcano Plot, white dots—unaltered; blue dots—downregulated; red dots—upregulated. Three independent experiments. Each plate represents a pool of samples (n = 3) from each independent experiment. (**B**) Table of downregulated miRNAs; (**C**) Table of upregulated miRNAs. (**D**) Interactome of modulated miRNAs and their predicted targets in hiNPCs from public RNA-sequencing.

**Figure 3 ijms-23-10282-f003:**
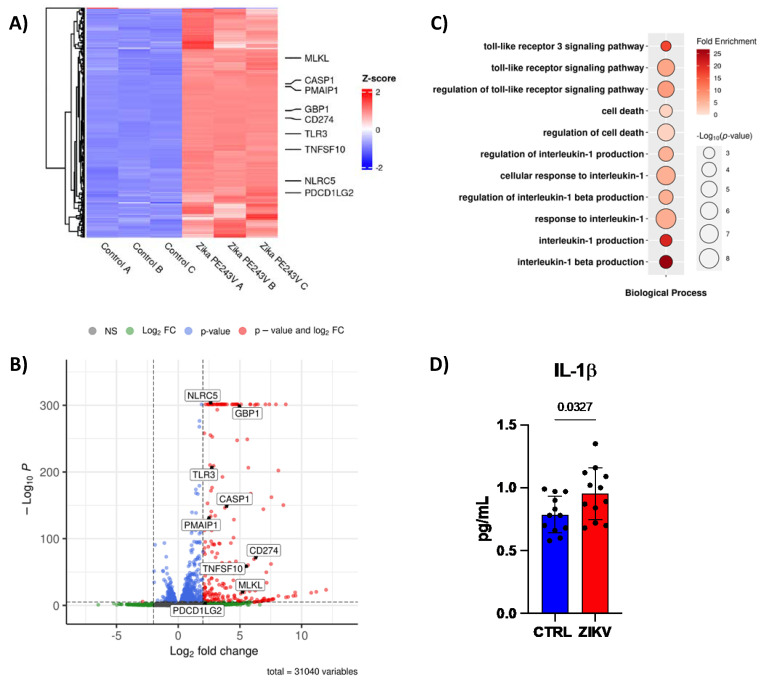
ZIKV downregulates miRNAs increasging cell death. (**A**) Heat-map, (**B**) bubble plot of gene ontology, and (**C**) volcano plot of public RNA-sequencing from hiNPCs (n = 3). 31,040 variables. (**D**) Quantification of IL−1β by CBA in supernatant of CTRL or ZIKV infected human NPCs after 48 h post infection (n = 12). Graphs represent three independent experiments. Unpaired *t*-test.

**Figure 4 ijms-23-10282-f004:**
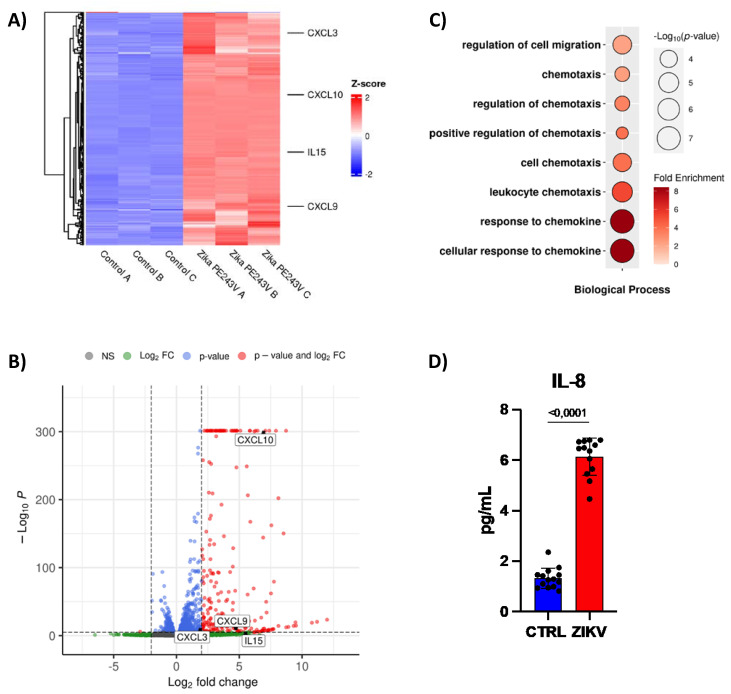
ZIKV downregulates miRNAs to recruit immune cells. (**A**) Heat-map, (**B**) Bubble plot of gene ontology, and (**C**) volcano plot of public RNA-sequencing from hiNPCs (n = 3). 31,040 variables. (**D**) Quantification of IL−8 by CBA in supernatant of CTRL or ZIKV infected human NPCs after 48 h post infection (n = 12). Graphs represent three independent experiments. Unpaired *t*-test.

**Figure 5 ijms-23-10282-f005:**
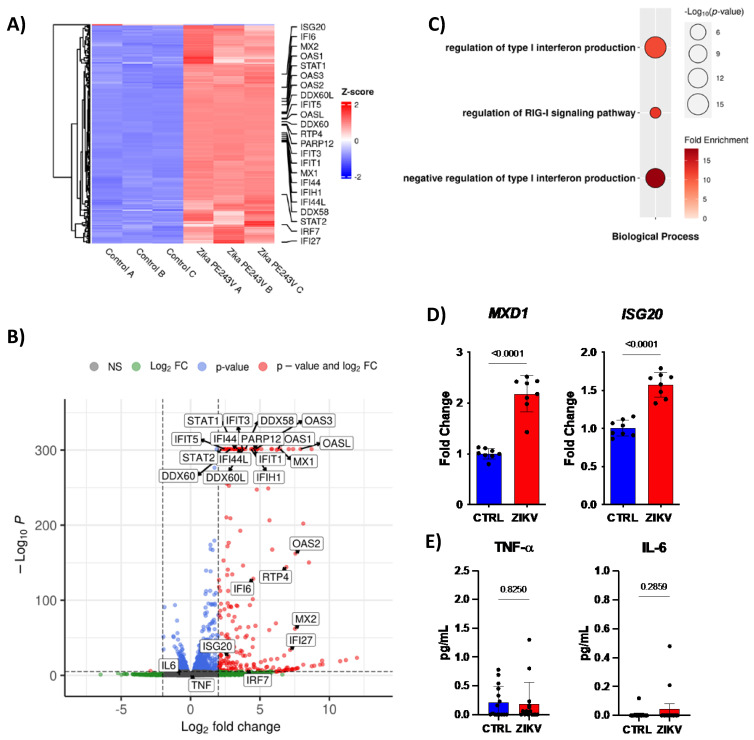
ZIKV downregulates miRNAs to modulate antiviral immune response. (**A**) Heat-map, (**B**) Bubble plot of gene ontology, and (**C**) volcano plot of public RNA-sequencing from hiNPCs (n = 3). 31,040 variables. (**D**) Gene expression of MXD1, and ISG20 by qPCR (n = 8), and (**E**) Quantification of TNF−α, and IL−6 by CBA in supernatant of CTRL or ZIKV infected human NPCs after 48 h post infection (n = 12). Graphs represent three independent experiments. Unpaired *t*-test.

**Figure 6 ijms-23-10282-f006:**
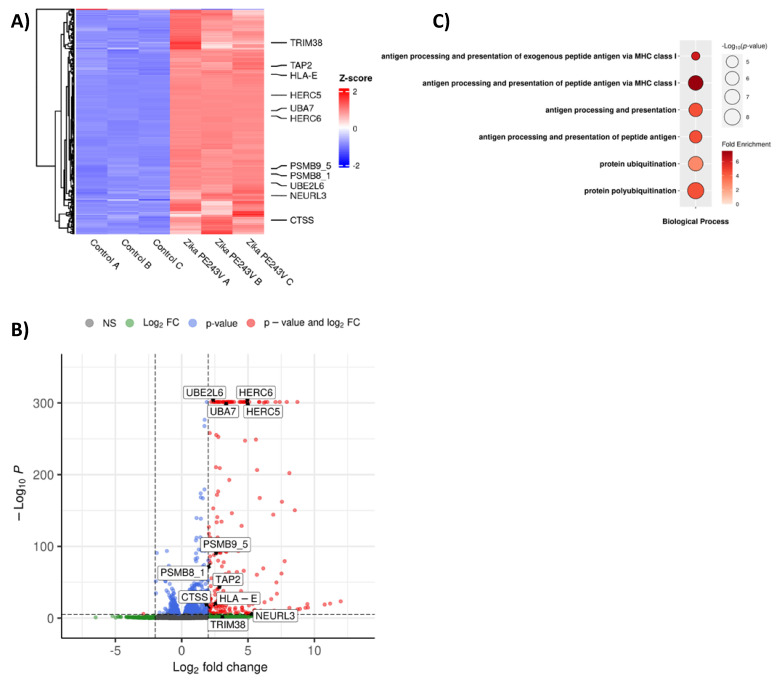
ZIKV downregulates miRNAs to increase ubiquitination and antigen presentation. (**A**) Heat-map, (**B**) Bubble plot of gene ontology, and (**C**) volcano plot of public RNA-sequencing from hiNPCs (n = 3). 31,040 variables.

**Table 1 ijms-23-10282-t001:** Primer sequence.

Gene	Forward Sequence	Reverse Sequence
** *ISG20* **	TCTACGACACGTCCACTGACA	CTGTTCTGGATGCTCTTGTGC
** *MXD1* **	CGTGGAGAGCACGGACTATC	CCAAGACACGCCTTGTGACT
** *HPRT* **	CCCTGGCGTCGTGATTAGTG	ACACCCTTTCCAAATCCTCAGC
** *ZIKV* **	TTGGTCATGATACTGCTGATTGC	CCTTCCACAAAGTCCCTATTGC

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
