# Peer review of "microRNAs Control Antiviral Immune Response, Cell Death and Chemotaxis Pathways in Human Neuronal Precursor Cells (NPCs) during Zika Virus Infection"

_ijms, 2022, doi:10.3390/ijms231810282_

Round 1
Reviewer 1 Report
The paper by Manganeli Polonio and colleagues is an experimantal study about ZIKV infecion of a neuronal stem cell model
The introduction is quite brief, and sufficiently explicative, but should be improved about miRNAs, as detailed in the minor points
The results section is a little bit too synthetic about the infection experiments. A single-time observation can demonstrate that the cells are infected, but
some more time points should be taken to better understand the behaviour of the virus in these cells (and, at least the T0 supernatants should be tested and compared with the
time point of interest).
The second part, about miRNAs discovery study is quite good. The third part, about this kind of meta-analysis on archive data, is well written, but in my view of relatively low interest.
There is nothing to demostrate that in the archive study the same miRNA expression profile was present, and the database predictions of miRNA targets are not so reliable.
Some experiments of miRNA target validation are required if the authors want to make full use of those predictions.
Or, alternatively, some in vitro expression analysis of the candidate target genes, to prove that they're differentially expressed in the experimental model obtained by the authors.
Only database predictions and archive datasets are not sufficient, in my humble opinion.
The discussion, despite being quite well written, is affected by the hybrid nature of the results, and draws conclusions only partially coming from actual data produced by the infection model,
and in a much larger part coming from the meta-analisis. I think it should be rewritten providing the reader with a better separation of the two 'kinds' of results and relative discussion.
The methods section is sufficiently informative, but should be improved, especially in the 'wet lab' protocols (see minor points)
The conclusion is in my opinion insufficiently informative.
Some minor points:
- please check for the aithors names, it looks like seomeone has uncorrect abbreviations
- In the abstract, line 8: 'posttranscriptionally'
- the keywords section is blank
- in the introduction, the section about miRNAs is not very informative, and the description of their role in physiopathological conditions is too
much simiplified. Moreover, the sentence beginning with 'other viruses' in unclear (which viruses?)
- the last period of the introduction is actually a conclusion. Please rewrite it or move it ot the appropriate section. Moreover, the last sentence is
badly written and should be corrected.
- In fig1, the A ) part is difficult to appreciate, due to small dimensions. Maybe a higher magnification of a smaller area could be more interesting
- In par 2.2, the cut-off for down/up regulation should be described (i.e.: 2 folds)
- In fig 2A the axes descritions are missing
- In fig 2D, the names of miRNAs are unreadable due to small dimensions.
- The last period of par 2.2 is incorrect: the data produced by this study don't demostrate this, while they support the data produced by the archive study.
The two studies taken together suggest a role for these miRNAs in ZIKV infection. Please clarify this concepts in the text.
- At the beginning of par. 4.1, 'and from the biorepository....'. Please rewrite in better form
- Paragraph 4.2 has some unclear part: There is the indication of the supplier of the viral isolate, but also of the supplier of the T3 stock. So, who
prepared the T1, T2 stocks?
- In par 4.3: Trizol doesn't extract mRNA, but total RNA.
- In par 4.4: the code of the ZIKV strain should be written also in par 4.2
- In par 4.5: the description of the extraction protocol is too long to be 'in brief', and nonetheless incomplete. Please rewrite more synthetically.
- Also in par 4.5: the components of molecular reactions should be detailed for concentration, not volume.
- In par 4.6: in the qPCR cycle description ,the number of seconds at 95°C is missing. Also, the description of DDCt could be wrtitten much better
- In par 4.10: please detail the viral protein specificity of the used anti-ZIKV primary antibody
- In par 4.11, please define, and briefly describe, the CBA assay
- references: please review the correct format of bibliographic references, there are several small errors (e.g. page numbers are often wrong)
- reference 2: please check for correct format
- reference 3: please check for correct format
- reference 6: please check for correct format
- reference 19: please check for correct format
- reference 35: please check for correct format
- reference 51: please check for correct format
Reviewer 2 Report
In this manuscript Polonio CM and co-authors elucidated the potential role of miRNAs in ZIKV infection of human neuroprogenitor cells (NPCs). Authors performed miRNA profiling of ZIKV-infected iPSC-derived NPCs by high-throughput qRT-PCR and identified three up-regulated and one down-regulated miRNAs. Authors then explored the publicly available RNA-Seq datasets of ZIKV-infected NPCs to identify the putative targets of these miRNAs. The predicted target mRNAs that exhibited inverse correlation in the expression level with ZIKV-affected mRNAs were further used for gene ontology enrichened analysis, which revealed their association with multiple biological processes related to immune response and cell death. This is an overly compelling study, however it has a limitation that, while not decreasing the value of the study, still needs to be acknowledged. In particular, authors should acknowledge that as miRNA activity can cause translational repression without degradation of target mRNAs, the transcriptomics data used in the study may not reflect the full spectrum of miRNA targets. In addition I have the following concerns that need to be addressed.
1. This study relies heavily of computational analysis, however the computational methods are described very briefly and don’t provide sufficient details to ensure reproducibility. In particular it is unclear which software was used for network reconstruction in Fig 2D and how this reconstruction was performed. For all other software used in data pre-processing, DEG and GO analysis authors should provide software versions and all settings.
2. The qRT-PCR cannot provide PFU equivalent (Fig 1B). It can provide gene copy numbers or relative abundance. Authors should change the axis labelling in Fig 1B accordingly.
Round 2
Reviewer 1 Report
The paper has been modified according to many of my recommendations, and I think it can be published in the current revised form
Author Response
"The paper has been modified according to many of my recommendations, and I think it can be published in the current revised form"
Thank you so much for all your contribution to improve this manuscript.